# A Review of Applications Using Mixed Materials of Cellulose, Nanocellulose and Carbon Nanotubes

**DOI:** 10.3390/nano10020186

**Published:** 2020-01-21

**Authors:** Daisuke Miyashiro, Ryo Hamano, Kazuo Umemura

**Affiliations:** 1Department of Physics, Tokyo University of Science, 1-3 Kagurazaka, Shinjuku-ku, Tokyo 162-8601, Japan; r.hamano.tus@gmail.com (R.H.); meicun2006@163.com (K.U.); 2ESTECH CORP., 2-7-31 Fukuura, Kanazawa-ku, Yokohama 236-0004, Japan

**Keywords:** cellulose, nanocellulose, carbon nanotubes, mixed materials

## Abstract

Carbon nanotubes (CNTs) have been extensively studied as one of the most interesting nanomaterials for over 25 years because they exhibit excellent mechanical, electrical, thermal, optical, and electrical properties. In the past decade, the number of publications and patents on cellulose and nanocellulose (NC) increased tenfold. Research on NC with excellent mechanical properties, flexibility, and transparency is accelerating due to the growing environmental problems surrounding us such as CO_2_ emissions, the accumulation of large amounts of plastic, and the depletion of energy resources such as oil. Research on mixed materials of cellulose, NC, and CNTs has been expanding because these materials exhibit various characteristics that can be controlled by varying the combination of cellulose, NC to CNTs while also being biodegradable and recyclable. An understanding of these mixed materials is required because these characteristics are diverse and are expected to solve various environmental problems. Thus far, many review papers on cellulose, NC or CNTs have been published. Although guidance for the suitable application of these mixed materials is necessary, there are few reviews summarizing them. Therefore, this review introduces the application and feature on mixed materials of cellulose, NC and CNTs.

## 1. Introduction

The environment around us is currently suffering from various problems such as CO_2_ emissions, the accumulation of large amounts of plastic, health damage due to air pollution, and the depletion of energy resources such as oil. In order to solve these problems, there is an increasing need for innovative recyclable materials, energy-saving materials, reinforcing materials, bio and chemical sensing materials, and portable electronic paper materials. Mixed materials of cellulose, nanocellulose (NC) and carbon nanotubes (CNTs) have great potential for such applications because these mixed materials can be adjusted to have various functions and structures due to the combination of cellulose and CNTs. However, there are various challenges in terms of the effectiveness, evaluation, appropriate use, production methods and cost for dissemination of these applications. Although guidance on the appropriate application and research of these mixed materials is necessary, there are currently few reviews summarizing them in a simple manner. Therefore, this review paper introduces the various applications and future prospects of mixed materials of cellulose and CNTs.

Many applications of CNTs have been examined, as they are one of the representative nanomaterials produced since the discovery of CNTs by Iijima in 1991 [1]. Over 25 years, many research and review papers have been actively published [2,3,4,5,6,7,8,9,10,11,12,13,14,15,16,17]. However, the application and commercialization of CNT-based products have been limited to luxury and expensive products [2,3] because the manufacturing process is complicated and high in cost [17,18,19]. This is mainly because CNTs easily aggregate in water. Aggregation not only causes a reduction in surface area and mechanical properties, but also causes the deterioration of various functions such as the optical and electrical properties. Thus, dispersion technology is important for the practical application of CNTs in a wide range of products, and various studies have been conducted to address this issue [10,11,12]. On the other hand, cellulose and NC, the main component of plant cell walls, wood is widely used and is the most abundant and renewable natural polymer on the earth. NC made from herbs, plants, wood, and organisms has been attracting attention because of its excellent features such as high strength, low thermal expansion, high aspect ratio, and light weight. In the last decade, many basic and applied research and review papers on cellulose and NC have been published [20,21,22,23,24,25,26,27,28,29,30], and many researchers have recognized the characteristics and application potential of them. The number of publications and patents related to cellulose and NC per year increased from 208 in 2009 to 2,372 in 2018 [30].

Recently, studies on mixed materials of cellulose, NC and CNTs have been actively conducted. In order to demonstrate the characteristics of these mixed materials and apply them appropriately, it is necessary to understand the characteristics of NC and CNTs themselves. The goal of this review is to provide a comprehensive understanding of the excellent individual properties of NC and CNTs, and the superior functions and applications that are possible when they are mixed. These mixed materials introduced in this paper include composites and hybrid materials. A composite represents a material in which the characteristics of the base material are enhanced by combining the base material with a dispersion material, and a hybrid material represents a new property different from the original by mixing different substances such as organic and inorganic substances at the molecular and atomic levels.

## 2. Feature of Nanocellulose and Carbon Nanotubes

In this section, we introduce the features of NC and CNTs such as their classifications, dimensions, mechanical, thermal, optical and electronical properties. A comparison of these characteristics will clarify the superiority and specificity of NC and CNTs.

### 2.1. Nanocellulose

Cellulose is the most abundant carbohydrate on earth. NC is a natural nanofiber that can be isolated by the defibration of cellulose from biomass resources such as wood, herbs, plants, and organisms. NC is roughly classified into cellulose nanofiber (CNF), cellulose nanocrystals (CNC), and bacterial nanocellulose (BNC).

CNF is a bundle of stretched cellulose nanofibers as shown in Figure 1a. The cellulose chains are entangled and flexible with a large surface area. Cellulose nanofibers, nanofibrillar cellulose, and cellulose nanofibrils are some of the synonyms for CNF. CNF is composed of amorphous regions, with widths of tens to several hundreds of nanometers, and soft and long chains of micrometer-scale length. CNF can have different mechanical properties depending on the natural source as shown in Table 1 [31,32]. CNC has a shape like an elongated crystal rod, as shown in Figure 1b, also called nanowhiskers, and has higher rigidity than NFC. CNC is usually 3 to 50 nm in width and 50 to 500 nm in length [33]. CNC has high axial rigidity (105–168 GPa), high Young’s modulus (20–50 GPa) [34], high tensile strength (about 9 GPa) [35], a low coefficient of thermal expansion (about 0.1 ppm/K) [36], high thermal stability (up to about 260 °C) [37], low density (1.5–1.6 g/cm^3^) [38], and thixotropy with time-dependent shear thinning properties [39]. BNC is synthesized and secreted by the Gluconoacetobacter xylinus family. Certain other bacterial species, such as *Agrobacterium, Pseudomonas, Rhizobium*, and *Sarcina*, also produce BNC. BNC is reported to have a high Young’s modulus (79–88 GPa) [40], high water retention capacity, and molecular weight up to 8000 Da [20]. Unlike CNC and CNF, which are prepared top-down by physical and chemical treatments, BNC can be prepared bottom-up from low molecular weight biomass by optimizing the culture conditions of cellulose-synthesizing bacteria.

### 2.2. Surface Modification of Nanocellulose

It is known that the easy aggregation of NC and CNTs is a common problem. So far, technologies for dispersing NC have been studied in various ways [41]. This section introduces some of the techniques for modifying the surface of NC. The surface modification technique is to impart ionic charges to NC. An outline of the carboxymethylation, phosphorylation, oxidation, and sulfonation processes is shown in Figure 2.

The carboxymethylation process introduces carboxymethyl groups onto the cellulose surface, making the surface negatively charged. There is a clear change in CNF generated from the carboxymethylation pathway. These suspensions are highly transparent and can be used in transparent films and composite applications. Phosphorylation that incorporates phosphate ester groups into the cellulose backbone significantly changes the original characteristics of cellulose. Thus, cellulose phosphorylation is a well-known surface modification technique for producing materials suitable for various fields of application such as biomedical applications, fibers, fuel cells, and so on. Oxidation mediated by 2,2,6,6-Tetramethylpiperidine-1-oxyl (TEMPO) can be used as a pretreatment to promote CNF separation and to render the NC surface hydrophobic. When TEMPO-NaOCl-NaOCl_2_ is used to catalytically oxidize cellulose, a high degree of charge is obtained due to the oxidation of the C6 position into anionic carboxylates, thus resulting in better dispersibility in water. The negative charges on the CNC surface introduced via this oxidation technique increase the electrostatic repulsive forces. Araki et al. reported the TEMPO-mediated oxidation of CNCs after hydrolyzing the cellulose fibers with HCl [42]. The rheological properties of TEMPO-oxidized CNCs have been studied [43]. Sulfonation is a technique for imparting an anionic charge to the surface of NC materials. Sulfuric acid hydrolysis that enables the formation of sulfate half esters from CNC hydroxyl groups provides a stable colloidal suspension [20].

### 2.3. Application of Nanocellulose

In this section, we introduce examples of NC that have been adopted for industrial applications. Controlling carbon dioxide emissions and reducing fuel consumption for the purpose of preventing global warming have been a long-standing issues in several fields such as the automotive industry, aviation, and railways. Most of the weight of these vehicles is in the body, so reducing this weight is important for reducing fuel consumption [44], but high rigidity of body is required in order to maintain safety and steering stability [45,46]. In order to address the issues of both fuel efficiency and safety, metal materials are being replaced with plastics and epoxy materials. For example, in a change in the 2000s, the midrange Audi A2 car adopted door trims made of polyurethane reinforced with a flax/sisal mixed material [47]. In addition, Toyota adopted a spare tire cover reinforced with kenaf fibers made of sugar cane and sweet potato with a PLA matrix in their RAUM 2003 model car. On the other hand, a concept car using NC was announced by the Finnish company UPM in 2015 [48]. More recently, a surprising nano cellulose vehicle (NCV) was exhibited at the Tokyo Motor Show 2019. NCV was realized through a supply chain through a consortium consisting of a total of 22 universities, research institutes with Kyoto University as the representative company. Composites made from plastic, epoxy and NC are used in various parts such as dashboards, door panels, and interior panels of vehicle body for reinforcements.

### 2.4. Carbon nanotubes

CNTs are one-dimensional tubular materials made of a wound graphene sheet composed of carbon atoms. There are single-wall carbon nanotubes (SWNTs), double-wall carbon nanotubes (DWNTs), and multi-wall carbon nanotubes (MWNTs). CNTs can have a radius ranging from 0.5 to 150 nm [49]. The mechanical properties vary between SWNTs, DWNTs, and MWNTs. In particular, CNTs have excellent mechanical properties such as a Young’s modulus of 70–1000 GPa [50,51,52], which is stronger, mass density 1.3–2.0 g/cm^3^ [53], thermoconductivity of 3000–3500 W/mK [54,55], coefficient of thermal expansion of 20.0 ppm/K [56] and capacitance 33–180 F/g [57,58]. Furthermore, SWNTs are known to exhibit metallic and semiconducting properties depending on how the graphene sheet composed of carbon atoms is wound, such as chirality [13,59]. SWNTs have unique optical properties such light absorption from visible region to infrared region and light emission in the near-infrared region (NIR) by absorbing the visible region. The light absorption spectrum of SWNTs can be divided into a metallic first transition (400–600 nm), a semiconducting first transition (600–900 nm), and a second transition (900–1600 nm) determined by chirality. It is expected to be applied as a probe for bio-imaging because a second transition (900–1600 nm), which is harmless to the human body, is hardly affected by the absorption and scattering of water and biomolecules.

The dispersion technology is important for applying CNTs to various applications, and there have been reports of dispersing CNTs with various organic substances and surfactants so far [10]. This techniques for CNTs are mainly divided into chemical modification and physical modification. Chemical modification refers to techniques in which functional groups that enable solvation are covalently introduced to the CNTs surface. Generally, CNTs surface oxidation by a strong acid treatment such as H_2_SO_4_/HNO_3_ is popular. However, chemical modification by covalent bonding may compromise the superior characteristics of CNTs because the modification breaks the bonds of the CNTs themselves. On the other hand, physical modification techniques can be divided into approaches that use hydrophobic interactions, such as micelle solubilization, and physical adsorption, such as π–π stacking. For example, various surfactants such as sodium dodecyl sulfate (SDS), sodium dodecylbenzene sulfate (SDBS), and sodium cholate (SC) are often used as micelle solubilizers [60]. In addition, the physical adsorption of organic molecules by π–π stacking has been actively studied. Among these, the composites of SWNTs wrapped with DNA developed in 2003 [61,62] are very stable, and have been studied as biomaterials for various applications such as biosensors and drug delivery [11,12].

### 2.5. Application of Carbon Nanotubes

CNTs have already been applied and commercialized for various purposes. For example, Mitsubishi Electric Corp. has adopted CNTs made by GSI Creos Corp. for the diaphragms of luxury in-vehicle speakers. For low-cost sensors that can communicate wirelessly, Andrews et al. demonstrated the use of fully printed carbon nanotube thin film transistors to detect environmental pressures in the pressure range of 0 to 42 PSIG [63]. Tyrata Inc. bought their technology license and is commercializing it as a tire sensor. As an application example of natural rubber, cross-linked natural rubber matrix mixed with MWNTs and conductive carbon black (CB) has been studied as conductive filler, and these elastic materials are expected to be used as sensors for various dynamic elastomer parts such as health monitoring, tires, valves, gaskets, and engine mounts [64]. Furthermore, supercapacitors using CNTs are studied for vehicle battery applications [65]. CNT sponges have been studied as a unique application to solves environmental problems [66]. This type of densified sponge expands instantly when it comes into contact with an organic solvent and is highly recyclable. In order to realize the above applications, high-cost CNTs have been limited to practical use. However, ZEON Corp. has begun operation of the world’s first mass production plant for CNTs using the Super-growt h method [59]. In the future, there are a variety of applications that are expected to be industrialized.

Mixed material of cellulose and CNTs that utilizes the advantages of each is expected to be used in various fields such as medical diagnosis, environment, automobiles, aviation, and precision electronics. The characteristics of cellulose and CNTs are summarized in Table 2.

## 3. Mixed Materials of Cellulose, Nanocellulose and Carbon Nanotubes

Mixed materials of cellulose, NC and CNTs have been studied for their dispersion performance, mechanical, optical, thermal and electronical properties and so on. These materials have been synthesized with various features and in various structures depending on the desired application. In this section, we introduce the various applications using these mixed materials.

### 3.1. Composites 

Dispersion technology is a very important key technology for realizing applications of composites of cellulose, NC and CNTs. Although we discussed the outstanding characteristics of NC and CNTs in Section 2, CNTs typically exist in a bundle state due to strong π stacking and van der Waals forces in the solid state. The high performance of these nanomaterials cannot be demonstrated when they are in an aggregated state. CNTs can be dispersed by ultrasonic irradiation in water or an organic solvent. However, the CNTs then form bundles again when the ultrasonic irradiation is stopped. Therefore, a dispersant is required to disperse CNTs in the solvent. 

Among the many types of cellulose, carboxymethyl cellulose (CMC) is a popular cellulose derivative and CMC has been used in a wide range of applications such as additives for food and feed, cosmetics, thickeners, water absorbents, and water retention agents. The reason that CMC has been in practical use for a long time is because it is non-toxic, biodegradable, biocompatible, and water-soluble. Thus, CMC is notable as a compatible dispersant for CNTs. Composites of SWNTs and MWNTs wrapped with CMC are expected to serve as an important platform for CNTs in optical, thermal and electronical applications and biomaterials since CMC can be safely mass-produced at low cost and recycled as a dispersant. It has been reported that CMC–SWNTs dispersed with CMC is nearly 20 times more stable than dispersion with conventional surfactants [67,68].

Regarding optical properties of CMC–SWNTs, Matsukawa et al. reported differences in the response of the NIR absorption spectra of CMC–SWNTs and DNA–SWNTs [69]. It is known that both CMC and DNA form π–π stacking and are adsorbed on the surface of SWNTs. However, Atomic force microscopy (AFM) observations show that the difference between CMC and DNA Cannot be distinguished because it is small at the nanoscale, as shown in Figure 3a,b. Remarkable NIR absorption peaks in the spectra of CMC–SWNTs and dsDNA–SWNTs appear at about 1130 nm and 1270 nm, as shown in Figure 3c,d, respectively. NIR absorption measurement results of the redox reaction of SWNT show a clear difference that the reaction of CMC–SWNTs is small and the reaction of DNA–SWNTs is large. The redox reactions of DNA–SWNTs have been studied by various groups [11,12]. For example, they found that peaks around 1270 nm in the NIR absorption spectrum of DNA-SWNT were reduced in the presence of hydrogen peroxide (H_2_O_2_), such as oxidizing agents. By contrast, in the presence of caffeine [70], dithiothreitol [71], β-mercaptoethanol and catechin [69] reducing agents the NIR peak was found to recover. In contrary, CMC–SWNTs exhibit an interesting behavior in that the redox reaction is small. 

CMC–SWNTs is used as a comparative material for sensor sensitivity studies. Oura et al. investigated the biomolecular recognition ability of RecA protein, which is well known to be able to recognize single strand DNA (ssDNA), in ssDNA–SWNTs using CMC–SWNTs as a comparison target [72]. They showed the difference the diameter of ssDNA-SWNTs with and without RecA protein. In contrast, the diameter change of CMC-SWNTs was small with or without RecA protein. From a structural approach, Riou et al. showed that CMC forms an apparently non-helical superstructure with CNTs, leading to their individualization [73]. In the future, as the mechanism governing the interactions between cellulose and CNTs becomes clearer, it is considered that this composite can be efficiently dispersed under optimum conditions as a reference material for sensors.

Composites of cellulose and CNT can be used as sensors that can detect certain chemical substances because their properties and sensitivity can be modified by changing the mixture ratio and material combination. For example, these chemical sensors include such diverse devices as concentrators for detecting for benzene, toluene, and xylenes [74], pH sensors made with titanium dioxide/multiwall carbon nanotube/cellulose hybrids [75]. In some cases, MWNT-g-CMC, in which CMC is grafted onto MWNTs using plasma technology, has a much higher adsorption capacity for removing UO_2_^2+^ than untreated MWNTs [76].

### 3.2. Aerogels

In recent years, aerogels of cellulose and CNTs have attracted much attention for their potential applications in many areas of basic research. These aerogels are considered promising materials for the development of supercapacitors for flexible energy storage [77,78,79,80,81], optical devices [82], thermoelectric devices [83], and adsorbents to address the problems arising from oil compound spills because these materials have excellent mechanical properties and electrochemical performance and they can selectively adsorb organic compounds. Several approaches have been proposed for manufacturing these aerogels [84,85,86]. For example, Xu et al. reported a new approach to producing a CNF–MWNT aerogel using bamboo powder of a raw material by a simple dipping and carbonization process. This CNF–MWNT aerogel can be recycled many times by distillation and combustion, meeting the requirements for practical oil-water separation [86]. 

One of the advantages of mixed materials is that various characteristics can be adjusted by changing the mass fractions of the component materials. Long et al. prepared aerogels with excellent mechanical properties using carboxymethylcellulose (CMC) as a raw material and carboxyl CNTs as reinforcements and investigated the surface morphology, specific surface area, compression modulus, density, and adsorption capacity of these materials for various oils, as shown in Figure 4. It was found that aerogels prepared with a mass fraction of CNTs exhibited an increased adsorption capacity for highly viscous liquids [84]. On the other hand, there has also been research on the deformation mechanism of aerogels in order to progress the material design of these aerogels. Hajian et al. investigated the potential of these CMC and CNT aerogels (CMC–CNT) to form superelastic and conductive aerogels for applications such as mechanically responsive materials through repeated conductivity tests at various strains. As a result, it has been found that the conductivity of aerogels of 50 wt % single SWNTs with a density of 10 kg/m^3^ and a porosity of 99% exceeds 0.5 S/cm [85]. CMC–CNT aerogels have been shown to have the potential to form recyclable superelastic aerogels without material loss or the need for other chemical treatments. Furthermore, Wang et al. showed that mixed materials of CNF cleaved from plant cell walls and functionalized MWNTs exhibited elastic mechanical behavior in combination with reversible electrical response under compression as well as responsive conductivity and pressure [87]. The synergistic combination of the wide availability of NC and the electrical functions of CNTs can lead to applications in devices such as supercapacitors and electrodes.

As research using other combinations of mixed materials, Zheng et al. prepared a polyvinyl alcohol (PVA), CNF, and MWNT mixed organic aerogel using a freeze-drying process. Adding small amounts of CNF and MWNTs significantly improved the mechanical properties of PVA aerogels, which showed an exponential dependence on relative aerogel density. These low-density aerogels have also been shown to exhibit very low thermal conductivity and high surface area, suggesting that they can be useful for many applications, including thermal insulation and structural components [78]. On the other hand, Yang et al. reported flexible electrodes of aerogels for supercapacitors fabricated by combining freeze-drying with a cold-press process to create a composite system of CNF, MWNTs, and polyaniline (PANI) after polymerizing polyaniline onto the surface of the CNF and CNTs by an in-situ polymerization method. These aerogels have a small charge transfer resistance (1.24 Ω), because PANI acts as a binder that binds tightly to each component, and a loose cross-section. Due to the 3D porous structure of the aerogel electrode, a high specific capacitance of 791.13 F/g was obtained at 0.2 A/g. In addition, the electrodes made from the aerogels showed excellent redox reversibility and cycle stability [79].

Zheng et al. used a new type of very flexible composite using H_2_SO_4_, poly vinyl alcohol (PVA) gel with CNF, reduced graphene oxide (RGO), to form a carbon nanotube hybrid (CNF–RGO–CNT) aerogel for use as an electrode material [81]. They used this aerogel to develop a solid supercapacitor electrolyte. These flexible solid supercapacitors were manufactured without a binder, current collector, or electroactive additive. Due to the porous structure of the CNF–RGO–CNT aerogel electrode and the excellent electrolyte absorption characteristics of CNF present in the aerogel electrode, the resulting flexible supercapacitors exhibited a high specific capacitance (252 F/g at a discharge current density of 0.5 A/g) and a remarkable cycle stability (more than 99.5% of the capacitance was retained after 1000 charge–discharge cycles at a current density of 1 A/g). Moreover, the supercapacitors also showed high areal capacitance, power density and energy density (216 mF/cm^2^, 9.5 mW/cm^2^, and 28.4 μWh/cm^2^, respectively).

Aerogels with excellent electrochemical performance and adsorption capacity can be used to fabricate low-cost, recyclable, high-performance supercapacitors and adsorbents, and are expected to be materials that can solve environmental and energy problems.

### 3.3. Smart Paper and Film

The research of smart paper and film technology is of great interest for portable electronics, sensor and shields applications. Mixed materials of cellulose and CNTs are expected to be used in low-cost, renewable smart paper that can replace petrochemical-based materials because of their excellent mechanical strength, lightweight and conductivity. The main features of mixed materials with CNTs on flexible and transparent cellulose paper are their mechanical strength, lightweight, and electrical conductivity. This section introduces paper and film applications realized with mixed materials of NC and CNTs. Major applications include electromagnetic interference shields [88,89,90], chemical sensors [91,92,93,94], conductive films [95,96,97,98,99], stable substrates [100,101], 3D printer ink bases [102,103], supercapacitors [104], and electrodes [105].

Smart paper is required to be flexible and have high mechanical strength and stable conductivity, and a filtration process capable of realizing paper that meets these requirements has been reported. Koga et al. coated cellulose nanofiber paper uniformly with conductive nanomaterials such as silver nanowires (AgNWs) and CNTs to produce cellulose nanopaper that functions as both a filter and a transparent flexible substrate [100]. AgNWs or CNTs filtration coating of on cellulose nanopaper was prepared by a two-step filtration process. First, an aqueous dispersion of cellulose nanofibers was filtered through a commercial membrane filter (pore diameter of 0.1 μm) for 20 min to form a wet nanopaper. Second, either the AgNW or the CNT aqueous dispersion was poured on the wet nanopaper, followed by filtration dewatering for 20 min. During the second filtration process, the wet nanopaper acted as an effective filter and retained almost all the conductive nanomaterials on the surface. Finally, the obtained material was completely dried by hot pressing at 110 °C for 20 min (1.1 MPa), and then peeled off from the membrane filter. The prepared AgNW@nanopapers and CNT@nanopapers showed the superiority of cellulose paper compared with AgNW@PET and CNT@PET made of polyethylene terephthalate film (PET). Furthermore, compared with cellulose nanopaper prepared by bar coating and spin coating, it was shown that the filtration coating method was simple and effective. The AgNW@nanopapers and CNT@nanopapers exhibited a sheet resistance of 12 Ω/ sq. and 88% light transmission. This is up to 75 times the sheet resistance of PET made by conventional coating processes. These results show that the filtration coating provides a conductive network that is uniformly connected to the vertical drainage through the paper-specific nanopores, whereas the conventional coating process is difficult, induces unavoidable self-aggregation, and the distribution of nanomaterial is uneven. In addition, the conductive network is embedded in the surface layer of the cellulose nanopapers, exhibits strong adhesion to the cellulose nanopaper substrate, provides foldability, and exhibits only negligible changes in conductivity. Therefore, this filtration process is expected to serve as an effective coating technique for a variety of conductive materials. 

These cellulose papers are expected to find applications as electromagnetic interference shields and chemical sensors. Electromagnetic interference shielding is an important application in modern communications and computer technology. Fugetsu et al. developed a composite material combining CNTs and cellulose paper [88]. CNTs form a continuous interconnected network on cellulose fibers, optimizing the conditions for mass production of paper. Papers provide an indispensable shielding mechanism that can shield electromagnetic interference, especially in the range of 30–40 GHz. These papers have been reported to be physically strong and flexible. On the other hand, Moilanen et al. produced CNT–CNC with a new material to replace traditional metal-based shields for EMI shielding. By combining a layered structure, including the CNT–CNC layer, with existing commercially available lossy materials (such as ferrite sheets), an effective EMI shield can be formed without degrading signal integrity performance [90]. It is worth noting that the shielding effect of the laminated CNT–CNC layer is greatly improved.

Cellulose paper not only has a large surface area but also provides the high foldability and twistability required for a robust and flexible sensor, making it effective for sensor applications. Han et al. demonstrated a humidity sensor by adding single-walled carbon nanotubes functionalized with carboxylic acid on cellulose paper. The conductance shift of the nanotube network intertwined with CNF is used for humidity sensing [91]. The sensor response was reported to be linear with good reproducibility and low hysteresis up to 75% relative humidity. In addition, the excellent adhesion of CNTs to cellulose paper shows excellent robustness against mechanical stresses such as bending, folding, and wrinkling, so water-based conductive CNT ink and cellulose paper are also expected to be applied to electrical circuits, chemical sensors, and cell scaffolds. Han et al. synthesized SWNTs with sodium dodecyl benzene sulfonate as a surfactant. Chemical sensors made by writing directly on cellulose paper showed good responses to up to 10 ppm ammonia vapor in air [92]. 

On the other hand, cellulose paper has good adsorptivity and compatibility with ink made from CNTs. There is research on conductive inks that are a composite of cellulose nanofibers and CNTs. Composite conductive ink made by mixing CNF and CNTs has had a great effect on cell research by imitating real nerve tissue more realistically. These inks have important applications for neural tissue engineering because they rely heavily on scaffolds that support the development of cells into functional tissues [102]. As other studies, Nguyen et al. developed a simple and environmentally friendly method to produce free-standing CNC–CNT films by preparing and depositing stable CNC–CNT dispersions [105]. The structural and morphological properties of CNC–CNT films carbonized at 800 °C were investigated by X-ray diffraction (XRD), Raman spectroscopy, and scanning electron microscopy (SEM). The conductive CNC matrix was generated by chemical carbonization. These carbonized composite films are expected to be used as anodes for lithium-ion batteries.

Many of the smart paper applications introduced so far have involved mixed materials of CNF and MWNTs. On the other hand, TEMPO–SWNTs using SWNTs, which are difficult to obtain as high-quality dispersions, such as mixed and industrialized TEMPO-oxidized NC are practical and attracting attention. TEMPO oxidized NC obtained by applying a TEMPO catalyst to wood fiber (pulp) and defibrating the resulting product mechanically showed strong mechanical properties, transparency, dispersibility, and unique viscosity behavior such as thixotropy [39]. As a composite with SWNTs, Koga et al. prepared TOCNs–SWNTs by mixing TEMPO-oxidized cellulose nanofibrils (TOCNs) with abundant sodium carboxyl groups on the crystalline nanocellulose surfaces [95]. The composite aqueous dispersion can impart conductivity to various substrate materials simply by coating and drying. TOCNs–SWNTs have been reported to form a flexible transparent conductive film with visible light transmittance of about 70% (including PET substrate) and sheet resistance of about 1.2 kΩ/γ. Furthermore, the effectiveness of TEMPO-oxidized cellulose was confirmed by the result that the resistance was increased to 5.8 kΩ/γ due to the inevitable aggregation of SWNTs without adding cellulose. In addition, these composites could be used as humidity sensors because the resistance of their one was confirmed to change in response to humidity. Thus, TOCNs–SWNTs can be expected to be applied as printable transparent conductive films and wearable electronic sensor devices [95] because they can be inkjet printed onto paper and can also be used to draw electrical wiring freely, as shown in Figure 5. Hamedi et al. also suggested that CNF and SWNTs can act as excellent aqueous dispersants, enabling low-cost exfoliation and purification of SWNTs with dispersion limits exceeding 40% by weight [96]. This dispersion method may be an inexpensive and sustainable alternative for the molecular self-assembly of advanced composite materials. They also investigated the characteristics of nanopaper, translucent conductive films, aerogels, and anisotropic microscale fibers produced from NFC–SWNTs. CNF–SWNT nanopapers exhibited a modulus of 13.3 GPa and 307 MPa when the strength increased at 3 wt % SWNTs.

Furthermore, Ito et al. produced a transparent film composed of SWNTs and a large amount of CMC and evaluated the effect of CMC wrapping on the photoluminescence (PL) characteristics. The PL spectrum from the transparent CMC–SWNT film showed a peak shift that depended on the SWNT type, and they verified that the optical properties of SWNTs were retained evenly across the film. The Raman scattering spectrum showed that SWMC was under uniaxial compression strain in the CMC film [97]. Smart paper and films using mixed materials of NC and CNTs are expected to have various applications that cannot be achieved with conventional paper and film.

### 3.4. Fibers

Highly conductive and mechanically strong microfibers are attractive for energy storage devices, thermal management, wearable electronics, and bioelectronic therapy. In this section, we introduce the applications of fibers and fiber mats that have excellent mechanical strength and electrical conductivity realized by mixing cellulose and CNTs. 

As a composite fiber manufacturing method, it is known that MWNTs in cellulose can be dissolved in ionic liquids and the fibers obtained by subsequent grinding and spinning show good dispersion and alignment [106]. Meanwhile, there have been research reports on the use of an electrospinning method for the dispersion of NC and CNTs [107,108,109]. The electrospinning method has the advantage that even a material that is difficult to process by conventional spinning or wet spinning can be easily made into a fiber and a nano-sized diameter can be obtained. Deng et al. prepared a composite by electrospinning a MWNTs and a cellulose acetate blend solution followed by deacetylation [107]. The effects of nanotubes on the resulting CNF precursor and microstructure stabilization were investigated using thermogravimetric analysis, TEM, and Raman spectroscopy. As a result, the embedded MWNTs were demonstrated to reduce the activation energy for the oxidative stabilization of cellulose nanofibers from about 230 to about 180 kJ/mol. In addition, Li et al. reported that they developed highly conductive CNF–CNT microfibers using high-speed and scalable 3D printing techniques [110]. These microfibers can be normally dispersed in an aqueous solution using TEMPO-oxidized CNF, resulting in a mixed solution with obvious shear thinning properties. Both CNF and CNT fibers in the fiber-based microfiber are well aligned, improving the interaction and penetration between these two building blocks, resulting in high mechanical strength (247 ± 5 MPa) and electrical conductivity (216.7 ± 10 S/cm).

On the other hand, compared to the CNF and MWNT fibers reported so far, although it is technically difficult to produce composites using CNF and SWNTs with complete characteristics, Wan et al. reported CNF–SWNT filaments composed of axial building blocks with a flexible CNF network. This filament exhibited high strength up to ~472.17 MPa and strain of ~11.77% shown in Figure 6a, surpassing most results reported for CNF–SWNT investigated previously in the literature as shown in Figure 6c. Interestingly, although the mechanical properties change depending on the CNTs ratio contained in the composites, the tendency of these properties changes greatly depending on the relative humidity, as shown in Figure 6a,b. These multifunctional filaments can be further manufactured as strain sensors that measure mass changes and investigate muscle movement and are expected to be used in the fields of portable gauge measurement and wearable bioelectronic therapy, as shown in Figure 6d [111]. Although young’s modulus of skeletal muscle myofibrils under physiological conditions has been studied to be several MPa in the axial direction and several KPa in the radial direction [112,113], these applications are higher elastic properties than those. Other unique devices such as non-woven macrofiber mats using CNF and SWNTs have also been reported by Niu et al. Features can be easily adjusted by controlling the extrusion pattern of the CNF–SWNT suspension in an ethanol coagulation bath and drying in air under limited conditions [114]. The new wearable supercapacitors based on non-woven macrofiber mats have been proven to have excellent adjustability, electrochemical stability, and damage reliability. Thus, these fiber and filament materials are expected to be applied as wearable electronics and biological signal sensor.

## 4. Mixed Materials of Bacterial Nanocellulose and Carbon Nanotubes

NFC and CNC made from plant and wood materials such as pulp can be top-down prepared by physical and chemical treatments. In contrast, bacterial nanocellulose (BNC) can be prepared bottom-up from low-molecular-weight biomass by optimizing the culture conditions for cellulose-synthesizing bacteria. Since BNC is produced from the bottom up, it has high uniformity and excellent fluidity, miscibility, and moldability. Thus, BNC is also expected to be applied to CNTs for functional enhancement. This section introduces various characteristics of BNC and its use in devices such as translucent conductive paper [115] capacitance aerogels [116] and conductive polymer films [117] for supercapacitors, as well as the excellent and selective absorption capacity for organic solvents and ability for high pressure detection, which are exhibited by BNC and carbon nanotubes.

The structure of BNC is affected by the strain, medium, and culture conditions used. Acid-treated MWNTs are added to a static medium, and their effects on bacterial cellulose structure are analyzed by SEM, AFM, and Fourier transform infrared spectroscopy (FT-IR). Further, CP/MAS ^13^C NMR, and X-ray diffraction measurements showed that the bacterial cellulose ribbon and MWNTs were intertwined to form three-dimensional network architecture, and even in the presence of MWNTs, a band-like assembly with sharp bending and rigidity was formed. It has been reported that these materials can be manufactured [118]. Taking advantage of the superior mechanical properties of BNC, Lee et al. developed recyclable and sensitive carbon nanotube resonators for chemical and biological applications. Conventionally, although sensitivity can be improved if the CNT size is small, there is still the problem of increased manufacturing complexity. This challenge can be overcome with new technologies that produce CNT-coated bacterial cellulose (BC) bundles that are long, sensitive, and high in tensile strength [119].

As applied research using BNC, Yoon et al. dispersed MWNTs in a surfactant (cationic cetyltrimethylammonium bromide) solution and soaked them in bacterial cellulose pellicle produced by Gluconacetobacter xylinum for 6, 12, and 24 h. Next, the surfactant was extracted with pure water and dried. Electron microscopy showed that individual MWNTs were strongly attached to the surface and interior of the cellulose pellicle [117]. These BNC and MWNT composites can be used as conducting polymer films, and the conductivity of this composite was found to be 1.4 × 10^−1^ S/cm based on the total cross-sectional area (about 9.6% MWNTs by weight). This result suggests that this composites creation process can not only disperse MWNTs in a network but also create conductive polymer films from them. In addition, Kang et al. achieved composites with high physical flexibility, desirable electrochemical properties, and excellent mechanical properties by rationally exploiting the interesting characteristics of BNC, CNTs, and ionic liquid-based polymer gel electrolytes. They demonstrated a complete all-solid flexible supercapacitor as shown in Figure 7. The supercapacitor performance of these composites remained high even after 200 bending cycles up to a radius of 3 mm. In addition, the supercapacitors showed excellent cycling with a Csp (~20 mF/cm^2^) decrease of only <0.5% after 5000 charge/discharge cycles at a current density of 10 A/g. These reports suggest the potential for BNC and CNTs to be an important basis for the development of flexible supercapacitors [120]. 

On the other hand, Hasan et al. modified BNC pellicles with MWNTs to develop flexible and conductive films capable of realizing a glucose biofuel cell system [121]. The membranes were further modified with redox enzymes such as pyroquinoline quinone glucose dehydrogenase (PQQ-GDH) and bilirubin oxidase (BODx), which function as anode and cathode catalysts using glucose as a biofuel source. The enzyme-functionalized MWNT cellulose-based glucose/O_2_ biofuel cell system utilizes the biochemical energy of glucose by using the oxidation of glucose and the reduction of molecular oxygen to generate power in the microwatt range. In relation to fuel cell application, Lv et al. experimentally showed that composites incorporating carboxylic acid multi-walled carbon nanotubes (c-MWNTs) in a BNC matrix have great potential for applications in renewable enzyme biofuel cells (EBFC). This EBFC biocathode and bioanode were prepared using BC/c-MWNT composites infused with laccase (Lac) and glucose oxidase (GOD) with the help of glutaraldehyde (GA) crosslinking. The electrochemical and biofuel performance of this composite was evaluated by cyclic voltammetry (CV) and linear sweep voltammetry (LSV). The EBFC power density and current density were 32.98 µW/cm^3^ and 0.29 mA/cm^3^, respectively. [122].

## 5. Prospects for mixed materials 

Up to this point, we have discussed the superior functions and applications of mixed materials of cellulose and CNTs. In this section, we discuss the future prospects of these nanomaterials. According to a MARKETS AND MARKETS survey, the market sizes of NC and CNTs are estimated to grow from US $240.7 million and US $3.95 billion in 2017 to US $661.3 million and US $9.84 billion by 2023, respectively [123,124]. Although one of the challenges for putting mixed materials into practical use is the manufacturing cost, the increase in the market size of NC and CNTs will further accelerate research on mixed materials and provide bright prospects for cost reduction. In the near future, these materials may replace body reinforcement materials such as automobiles, aircraft and industrial machinery. However, it is necessary to overcome some problems such as cost, durability, and safety when compared with conventional materials such as aluminum, epoxy resin, carbon fiber reinforced plastic (CFRP), and to explore the ways to make use of characteristics such as electrical properties, transparency, and recyclability that cannot be reproduced by conventional materials.

In order to overcome these problems, mixed materials are expected to be designed using simulation techniques to drive these applications efficiently and optimally. So far, simulation research on NC and CNTs have reported intermolecular forces and binding energies using molecular dynamics methods and structural analysis using finite element methods [125,126,127,128,129,130,131,132,133,134,135]. In addition, there are also reports of simulations on mixed materials of NC and CNTs [136,137]. In recent years, as a result of remarkable improvements in computer performance, many researches on NC or CNTs using machine learning have been reported [138,139,140,141,142,143,144]. In the future, research using these machine learning methods will accelerate the development of further mixed materials of cellulose and CNTs research, such as by reducing the time and cost of experiments.

## 6. Summary

Mixed materials of cellulose and CNTs can have various functions and take many forms, such as composites, aerogels, papers, films, or fibers depending on their combinations, as shown in Table 3. These mixed materials not only exhibit the excellent characteristics of cellulose, NC, and CNTs, but are also renewable, biodegradable, recyclable, and energy saving. In the next decade, the further development and spread of mixed materials will be required, as they are expected to be adopted in a variety of fields such as medical diagnostics, machinery, the automotive industry, and chemistry for a sustainable industrial society, as shown in Figure 8. This review will contribute to a comprehensive understanding of the characteristics, progress of these mixed materials and to overcoming the challenges presented by them.

## Figures and Tables

**Figure 1 nanomaterials-10-00186-f001:**
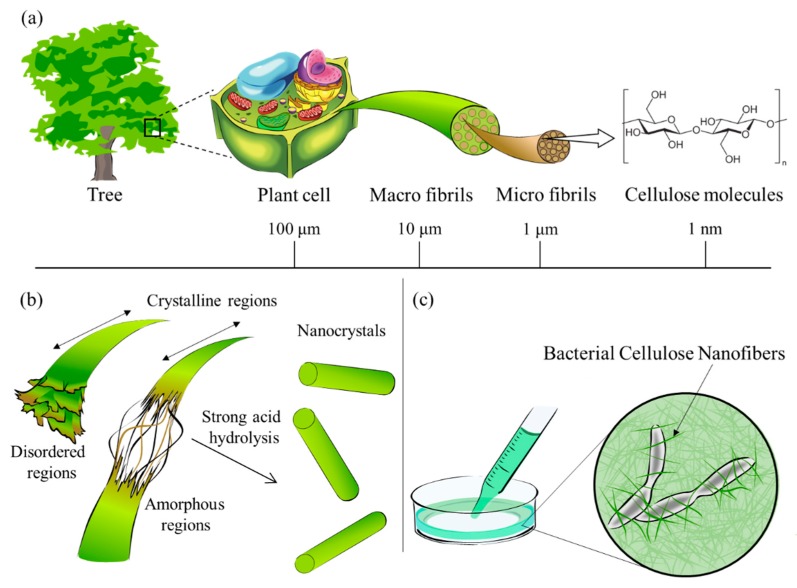
Cellulose contained in plants or trees has a hierarchical structure from the meter to the nanometer scale, as shown in (**a**). A schematic diagram of the reaction between cellulose and strong acid to obtain Nanocellulose is shown in (**b**). Bionanocellulose cultured from cellulose-synthesizing bacteria is shown in (**c**). Figure 1 was drawn by co-author R. Hamano.

**Figure 2 nanomaterials-10-00186-f002:**
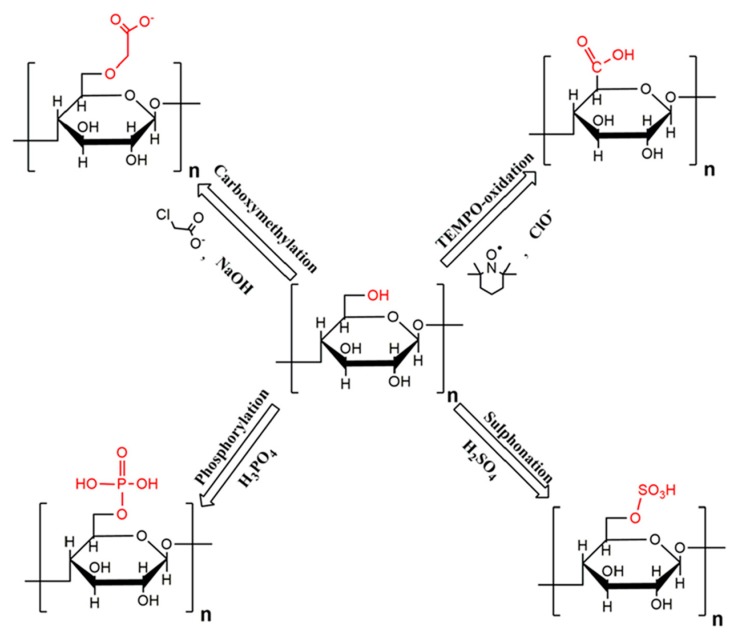
Different surface modification techniques through which ionic charges are imparted to the NC surface. Reproduced with permission from [20]. Copyright American Chemical Society, 2018.

**Figure 3 nanomaterials-10-00186-f003:**
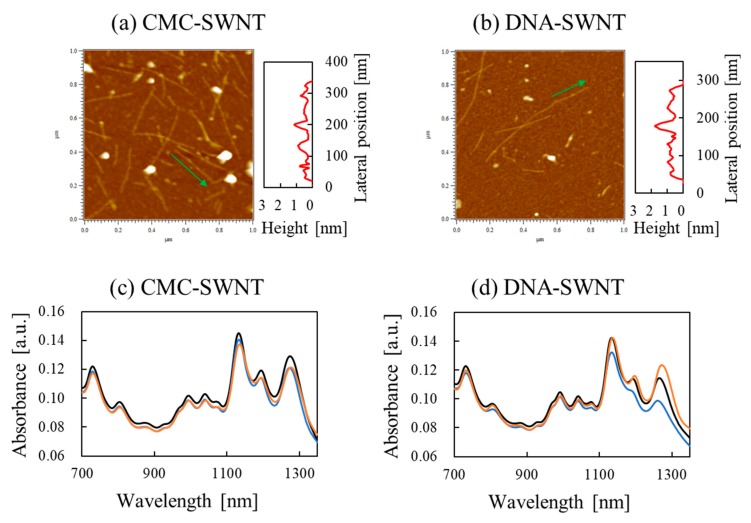
AFM images: (**a**) CMC–SWNTs, (**b**) dsDNA–SWNTs. NIR absorption spectra of SWNT composites: (**c**) CMC–SWNTs, (**d**) dsDNA–SWNTs. Black line: SWNT composites initially prior to adding H_2_O_2_. Blue line: SWNT composites oxidized with 0.03% H_2_O_2_ solution for 30 min. Orange line: the oxidized SWNT composites were incubated with 15 μg/mL catechin solution for 10 min. The absorption values represent the average of five independent measurements. The results of Figure 3 were measured by co-author R. Hamano.

**Figure 4 nanomaterials-10-00186-f004:**
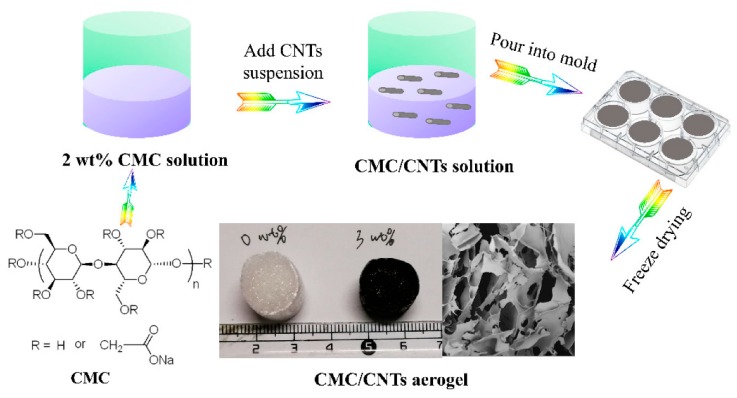
Preparation of CMC/CNTs aerogel. Reproduced with permission from [84]. Copyright MDPI, 2019.

**Figure 5 nanomaterials-10-00186-f005:**
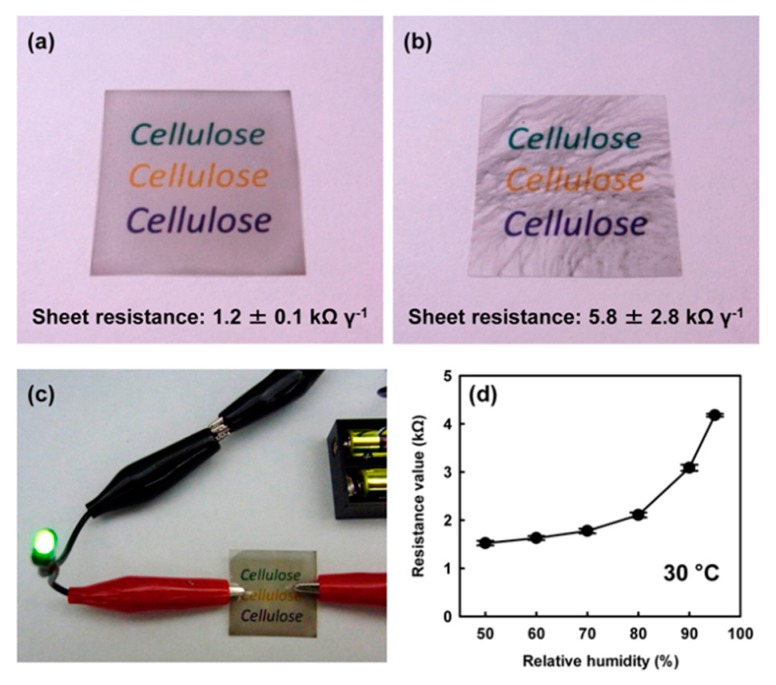
Optical images and sheet resistance measurements at 23 °C and 50% relative humidity (RH) for (**a**) TOCNs–SWNTs (**b**) the CNT cast onto PET films.; these films of 3 × 3 cm were laid on the color-printed word “cellulose” on copy paper. (**c**) The lighting of an LED using a transparent conductive film based on the TOCNs–SWNTs cast PET. (**d**) Correlation between the resistance values of the TOCNs–SWNTs-cast PET film at 30 °C, and the relative humidity. Reproduced with permission from [95]. Copyright American Chemical Society, 2013.

**Figure 6 nanomaterials-10-00186-f006:**
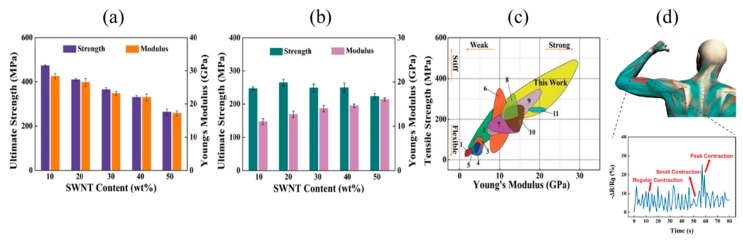
Mechanical properties of the filaments (**a**) Young’s modulus and ultimate strength of five different filaments investigated in this work at 12% relative humidity (RH). (**b**) Young’s modulus and ultimate strength of five different filaments investigated in this work at 50% RH. (**c**) Tensile stress and Young’s modulus map to compare different types of CNF/CNT composites. (1) CNF/MWNT films, (2) CNF/MWNT nanopapers, (3) CNF/fluorinated CNT films, (4) CNF/MWNT films, (5) CNF/SWNT films, (6) CNF/SWNT films, (7) CNF/SWNT membranes, (8) CNF/SWNT membranes, (9) CNF/MWNT nanopapers, (10) CNF/MWNT microfibers, and (11) CNF/SWNT microfibers. (**d**) Application of sensor that can record slight changes in electrical resistance and respond to muscular movement. Reproduced with permission from [111]. Copyright American Chemical Society, 2019.

**Figure 7 nanomaterials-10-00186-f007:**
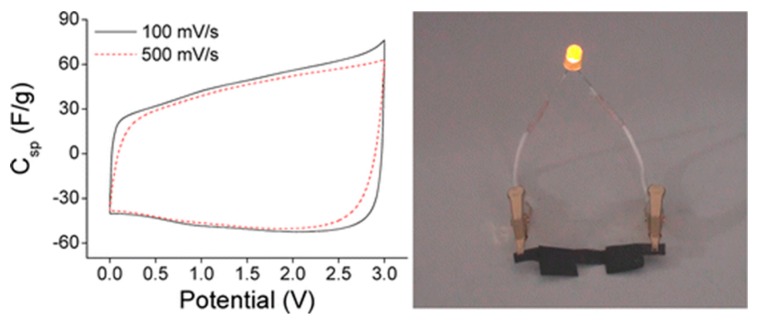
Characteristics of CNT/BNC/ion gel-based all-state flexible supercapacitors. The left figure shows cyclic voltammetry (CV) curves measured at scan rates of 100 and 500 mV/s. The right figure shows a photograph of a light-emitting diode (LED) turned on by the flexible supercapacitors. Reproduced with permission from [120]. Copyright American Chemical Society, 2013.

**Figure 8 nanomaterials-10-00186-f008:**
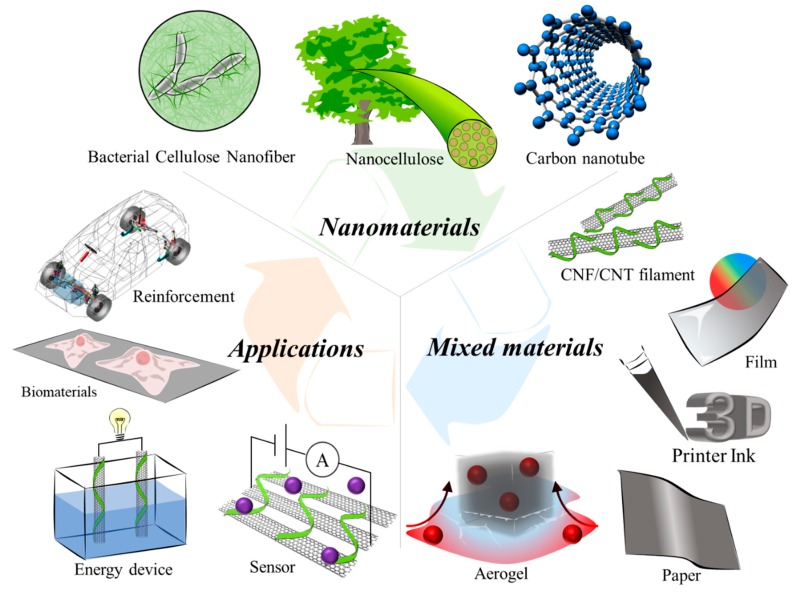
Diagram showing recycling from mixed materials of cellulose, nanocellulose and CNTs to applications. Figure 8 was drawn by author D. Miyashiro and co-author R. Hamano.

**Table 1 nanomaterials-10-00186-t001:** Comparison of mechanical properties of natural fibers [31].

	Tensile Strength [GPa]	Young’s Modulus [GPa]
Cotton	0.3–0.7	5.0–10.9
Wool	0.1–0.2	2.3–3.4
Silk	0.3–0.5	7.3–11.2
Flex	0.3–0.9	24.0
Jute	0.3–0.7	43.8
Sisa	0.4–0.6	–
Ramie	0.3–0.8	53.4

**Table 2 nanomaterials-10-00186-t002:** Comparison of NC and CNTs.

	CNC	CNF	BNC	Ref.	SWNT	MWNT	Ref.
Width [nm]	3–50	4–100	20–140	[33,40]	0.5–10	5–100	[49,53]
Length [nm]	100–500	5000–	5000–	[33,40]	10–1000	100–	[49,53]
Young’s modulus [GPa]	20–50	0.5–10	79–88	[31,34,40]	1000	70–950	[51,52,53]
Tensile strength [GPa]	9	0.1–1.0	21	[31,35,40]	13–53	11–150	[51,52,53]
Mass density [g/cm^3^]	1.5–1.6	1.3–1.4	1.1	[38,40]	1.3–1.5	1.8–2.0	[50]
Thermo conductivity [W/mK]	–	–	–	–	3500	3000	[54,55]
Coefficient of Thermal expansion [ppm/K]	0.1			[36]	20.0		[56]
Capacitance [F/g]	–	–	–	–	180	32.7	[57,58]
Remarks	Transparency thermal stability (±260 °C) [37] Thixotropic				ChiralityHigh dispersibility	Wide range ofdimensions	–

**Table 3 nanomaterials-10-00186-t003:** Applications created by the combination of mixed cellulose and CNTs.

Type	Materials			Purpose	Ref.
	Cellulose	Carbon Nanotube	Other		
Composites	Cellulose	SWNT	–	Detection of benzene, toluene, and xylene (BTX) vapors	[74]
Composites	Cellulose	MWNT	TiO_2_	pH sensor	[75]
Composites	CMC	MWNT	–	Removal sensor of UO_2_^2+^	[76]
Aerogels	CNF	MWNT	–	Flexible supercapacitors	[77]
Aerogels	CNF	MWNT	PVA	Application for insulation and structural parts	[78]
Aerogels	CNF	MWNT	PANI	Supercapacitance	[79]
Aerogels	CNF	MWNT	PVA	Adsorption material for organic solvent, oil	[80]
Aerogels	CNF	CNT	RGO	Super capacitor electrolyte	[81]
Aerogels	CMC	CNT	–	Conductive aerogels	[85]
Aerogels	CNF	MWNT	–	Sensor for detecting the conductivity and pressure	[87]
Aerogels	BNC	CNT	–	Adsorption material for organic solvent, oil	[116]
Aerogels	BNC	CNT	GA	Enzyme biofuel cell	[122]
Paper	Cellulose paper	CNT	–	EMI shielding	[88]
Paper	CNF	CNT	–	EMI shielding	[89]
Paper	CNC	CNT	–	EMI shielding	[90]
Paper	Cellulose paper	CNT	–	Humidity sensor	[91]
Paper	Cellulose paper	CNT	–	Ammonia sensor	[92]
Paper	Cellulose paper	MWNT	–	Chemical vapor sensor	[93]
paper	Cellulose paper	CNT	–	Smart paper for portable electronics and sensing	[94]
paper	CNF	CNT	–	Conductive paper	[99]
paper	Cellulose paper	SWNT	–	Transparent, Conductive paper	[100]
Paper	CNF	CNT	–	conductive inks for 3D printing of scaffolding	[102]
Paper	Cellulose paper	CNT	–	chemical sensor	[103]
Paper	CNF	CNT	LTO	Lithium ion battery (LIB) electrode	[104]
Paper	BNC	MWNT	–	Translucent conductive paper	[115]
Film	TOCNs	CNT	–	Highly conductive and printable nanocomposites	[95]
Film	CNF	SWNT	–	Translucent conductive film	[96]
Film	CMC	SWNT	–	Translucent conductive film	[97]
Film	CMC	CNT	GO	Reinforced film	[101]
Film	CNF	CNT	–	Lithium ion battery anode	[105]
Film	BNC	MWNT	–	Conductive polymer film	[117]
Film	BNC	CNT	–	Flexible supercapacitor	[120]
Film	BNC	MWNT	–	Flexible supercapacitor for biofuel cell	[121]
Fiber	CNF	MWNT	–	Super capacitor electrode	[107]
Fiber	CNF	MWNT	–	Conductive fiber mat	[108]
Fiber	CNF	MWNT	–	Reinforced fiber	[109]
Fiber	TEMPO-CNF	MWNT	–	Wearable electronic devices	[110]
Fiber	CNF	SWNT	–	Flexible strain and mass sensor	[111]
Fiber	CNF	SWNT	–	Wearable supercapacitor	[114]
Fiber	BNC	MWNT	–	Resonator for biochemical sensing	[119]

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
