# Peer review of "A Review of Applications Using Mixed Materials of Cellulose, Nanocellulose and Carbon Nanotubes"

_nanomaterials, 2020, doi:10.3390/nano10020186_

Round 1

Reviewer 1 Report

The article in object is a review concerning the applications of hybrid materials obtained from nanocellulose and carbon nanotubes. the topic is interesting and updated and the review, in my opinion, can be accepted after a major revision. Some issue to be addressed:

1: generally the authors are even too emphatic about the applications of these materials. The subject is indeed interesting but much work is still to be done to prove their utility.

2: When I read nanocellulose in the title I expected to find applications of nano forms of cellulose. As a matter of fact, the authors, on page 2, describe the nature of the different form of nanocellulose and their dimensions; however, when moving to describe the applications of the hybrids, the authors describes also examples with carboxymethylcellulose (CMC) in which CMC molecules (the polymer) are wrapped around carbon nanotubes (page 7). That is not an hybrid with nanocellulose, it is a composite between cellulose and CNT. In fact, the authors, in the first section of the application, do never states which kind on nanocellulose is used. It can be interesting to discuss these examples but then, please, make the title more general

3: I do not see the scope for describing, for the millionth time, the nature of CNT. Furthermore, the authors try to resume in a small chapter too much complex information and for this reason they make some mistakes. I do not agree that figure 3a shows an unrolled CNT: it is a 2D Bravais lattice in which the primitive vectors can be used to obtain the Chiral vector using the N and M indexes. However there is no need to report this information. If the reader do not know this, he can be redirected to the proper description using a reference. Section 2.5 is not comprehensive, despite what it is said at the beginning. the physical absorption on CNT surface does not occur through p-p bond but through p-p interactions or p-p-stacking (please correct it through the whole article, the p-bond is something different). I would personally reduce to a minimum sections 2.4, 2.5 and 2.6.

4: description  from line 214 to 237 is confusing. please reorganize it

5: line 251. toluene and xylenes

6: lines from 339 to 345: the authors talks about a filtration techniques that they have not discussed. I would suggest to treat this part in greater detail

7: line 392 to 394. please rewrite the sentences to make them clearer.

Author Response

Response to reviewer #1:

We are grateful for your precise comments and useful suggestions that have helped us to improve our paper. According to your comments, we modified the manuscript as follows:

1: generally the authors are even too emphatic about the applications of these materials. The subject is indeed interesting but much work is still to be done to prove their utility.

We agree with your comments and added the problem of the mixed material.
>>see from p1. line 33 to p1. line 35
>>see from p14. line 537 to p14. line 543

2: When I read nanocellulose in the title I expected to find applications of nano forms of cellulose. As a matter of fact, the authors, on page 2, describe the nature of the different form of nanocellulose and their dimensions; however, when moving to describe the applications of the hybrids, the authors describes also examples with carboxymethylcellulose (CMC) in which CMC molecules (the polymer) are wrapped around carbon nanotubes (page 7). That is not an hybrid with nanocellulose, it is a composite between cellulose and CNT. In fact, the authors, in the first section of the application, do never states which kind on nanocellulose is used. It can be interesting to discuss these examples but then, please, make the title more general

We have revised the “title” to make the our paper accurate.
>>see title,
“A review of applications using mixed materials of cellulose, nanocellulose and carbon nanotubes”
We clarify "hybrid materials" and "composites" to avoid confusion.
>>see from p2. line 61 to p2. line 65

3: I do not see the scope for describing, for the millionth time, the nature of CNT. Furthermore, the authors try to resume in a small chapter too much complex information and for this reason they make some mistakes. I do not agree that figure 3a shows an unrolled CNT: it is a 2D Bravais lattice in which the primitive vectors can be used to obtain the Chiral vector using the N and M indexes. However there is no need to report this information. If the reader do not know this, he can be redirected to the proper description using a reference. Section 2.5 is not comprehensive, despite what it is said at the beginning. the physical absorption on CNT surface does not occur through p-p bond but through p-p interactions or p-p-stacking (please correct it through the whole article, the p-bond is something different). I would personally reduce to a minimum sections 2.4, 2.5 and 2.6.

We deleted previous Figure 3 and tried to simplify it by reducing the section from 2.4 to 2.6.
>>see from p5. line 150 to p5. line 158
The “p-p bond” has been modified to “p-p stacking”.

4: description from line 214 to 237 is confusing. please reorganize it

The explanation of section 4 has been rearranged and simplified.

>>see from p7. line 218 to p7. line 231

5: line 251. toluene and xylenes

We fixed the word.

>>see from p7. line 245

6: lines from 339 to 345: the authors talks about a filtration techniques that they have not discussed. I would suggest to treat this part in greater detail

We have tried to elaborate on a filtration techniques.
>>see from p9. line 329 to p10. line 341

7: line 392 to 394. please rewrite the sentences to make them clearer.

We try to rewrite the sentences to make composites clearer.

>>see from p11. line 391 to p11. line 393

Reviewer 2 Report

This is a very important and timely review paper on nanocellulose and carbon nanotubes. The manuscript is nicely done. I only had some minor issues to report:

1) the authors should clarify "hybrid material" and "composite material". I am assuming they are not the same and composite material has a well defined definition. what is "hybrid material". The authors should have defined this terminology at the beginning of this review to avoid confusions.

2) In several occasions, the authors refer "section" in the paper as "chapter" . This is confusing. Please change to "sections"

overall, well done.    

Author Response

Response to reviewer #2:

We greatly appreciate your comments and evaluations. We brushed up the some sections without changing the essential conclusions as shown below.

1) the authors should clarify "hybrid material" and "composite material". I am assuming they are not the same and composite material has a well defined definition. what is "hybrid material". The authors should have defined this terminology at the beginning of this review to avoid confusions.

We clarify "hybrid materials" and "composite materials" to avoid confusion.
>>see from p2. line 61 to p2. line 65
We have revised the “title” to make the our paper accurate.
>>see title,

2) In several occasions, the authors refer "section" in the paper as "chapter" . This is confusing. Please change to "sections"

The “Chapter” has been modified to “sections”.

Round 2

Reviewer 1 Report

Authors improved the manuscript as requested. the article, in my opinion, can be accepted in the present form